# In Vitro and In Silico Screening and Characterization of Antimicrobial Napin Bioactive Protein in *Brassica juncea* and *Moringa oleifera*

**DOI:** 10.3390/molecules26072080

**Published:** 2021-04-05

**Authors:** Sangeeta Chandrashekar, Raman Vijayakumar, Ramachandran Chelliah, Eric Banan-Mwine Daliri, Inamul Hasan Madar, Ghazala Sultan, Momna Rubab, Fazle Elahi, Su-Jung Yeon, Deog-Hwan Oh

**Affiliations:** 1Department of Food Science and Biotechnology, College of Agriculture and Life Science, Kangwon National University, Chuncheon 24341, Korea; gsangeetakv@gmail.com (S.C.); ericdaliri@yahoo.com (E.B.-M.D.); rubab.momna@gmail.com (M.R.); elahidr@gmail.com (F.E.); sujung0811@gmail.com (S.-J.Y.); 2Department of Physiology, Bharath Institute of Higher Education and Research, Chennai 600073, India; 3Department of Biotechnology, School of Biotechnology and Genetic Engineering, Bharathidasan University, Tiruchirappalli 620024, India; inambioinfo@gmail.com; 4Department of Computer Science, Aligarh Muslim University, Aligarh 202002, India; ghazala.sultan2k17@gmail.com; 5School of Food and Agricultural Sciences, University of Management and Technology, Lahore 54770, Pakistan

**Keywords:** *Moringa oleifera*, *Brassica juncea*, coagulant protein, cell aggregation, growth inhibition, LCMS, napin, lipoprotein, topoisomerase, molecular-docking

## Abstract

The study aimed to investigate the antibacterial activity of Mustard (*Brassica juncea*) and Moringa (*Moringa oleifera*) leaf extracts and coagulant protein for their potential application in water treatment. Bacterial cell aggregation and growth kinetics studies were employed for thirteen bacterial strains with different concentrations of leaf extracts and coagulant protein. *Moringa oleifera* leaf extract (MOS) and coagulant protein showed cell aggregation against ten bacterial strains, whereas leaf extract alone showed growth inhibition of five bacterial strains for up to 6 h and five bacterial strains for up to 3 h. *Brassica juncea* leaf extract (BJS) showed growth inhibition for up to 6 h, and three bacterial strains showed inhibition for up to 3 h. The highest inhibition concentration with 2.5 mg/mL was 19 mm, and furthermore, the minimum inhibitory concentration (MIC) (0.5 mg/mL) and MBC (1.5 mg/mL) were determined to have a higher antibacterial effect for <3 KDa peptides. Based on LCMS analysis, napin was identified in both MOS and BJS; furthermore, the mode of action of napin peptide was determined on lipoprotein X complex (LpxC) and four-chained structured binding protein of bacterial type II topoisomerase (4PLB). The docking analysis has exhibited moderate to potent inhibition with a range of dock score −912.9 Kcal/mol. Thus, it possesses antibacterial-coagulant potential bioactive peptides present in the *Moringa oleifera* purified protein (MOP) and *Brassica juncea* purified protein (BJP) that could act as an effective antimicrobial agent to replace currently available antibiotics. The result implies that MOP and *Brassica juncea* purified coagulant (BJP) proteins may perform a wide degree of antibacterial functions against different pathogens.

## 1. Introduction

Water scarcity has been predicted globally and particularly in India by 2020 [1]. The major problem is poor water quality, and it is estimated that waterborne diseases affect about 37.7 million Indians annually and 1.5 million children under the age of five die due to diarrhea each year [2]. A major portion of the rural population depends mostly on groundwater, whereas the urban population depends on surface water. Increased pressure on the resources due to the alarmingly growing population and other factors such as industrial discharge, agricultural run-off, and poor sanitation practices put the long-term availability and quality of the potable water at stake.

In most developing countries, including India, farming by-products, crop residues, and grazing along with some protein and energy supplements are the main source of livestock feed for ruminant animals. Common protein supplements for ruminants are oilseed cakes obtained as oil industry by-products. Mustard (*Brassica juncea*) and Moringa (*Moringa oleifera*) cake is the most widely available protein substitute for livestock in Asian countries [3]. Microbial contamination through fecal contamination in water is the major reason for the poor water quality in developing countries, transmitting a large number of diseases. The common pathogens present in the drinking water include *Shigella species*, *Salmonella species*, *Klebsiella species*, *Escherichia coli*, *Enterobacter species*, and parasites such as *Giardia lamblia* and *Entamaebahistolytica* [4,5].

Drinking water treatment involves several combined processes based on the quality of the water source and the cost and availability of chemicals in achieving the desired level of treatment and water quality standard as recommended by the World Health Organization (WHO) [6]. The available water treatment processes are expensive, especially in developing countries. Synthetic organic and inorganic chemicals are commonly used for various water treatment processes and are associated with environmental and human health problems [7].

As an alternative, natural materials (plant material) can be used for water treatment. Moringa (*Moringa oleifera*) is one of the widely cultivated species in the tropical region of Asia, Africa, and South America, and it is used in rural areas of Africa for water treatment [8]. The previous study indicates that various parts of Moringa such as leaves, roots, and bark have antibacterial activity, and the seeds are well known for their coagulant and antibacterial properties [9,10,11,12,13,14,15]. The seed extract and recombinant protein of *M. oleifera* are effective against *Staphylococcus aureus*, *Streptococcus pyrogenes*, *Streptococcus mitis*, *Streptococcus pneumoniae*, *Enterococcus faecalis*, *Escherichia coli,* and *Legionella pneumophila* for up to 150 min [15]. Although *Moringa* leaf extract has been extensively studied to be a potent coagulant and antimicrobial agent, there are some concerns such as (i) the availability of *Moringa* leaf and cost, (ii) the difference in the coagulation property of leaf extracts collected from different localities, and (iii) the type and quality of surface water (presence of contaminants, physical and chemical properties) to be treated. A screening study was conducted to find out coagulant protein from plant materials as a complement to *Moringa* seeds for water treatment in Southern India [16]. 

The Yellow (*Sinapis alba*) and Brown (*Brassica juncea*) mustards possess glucosinolates. These glucosinolates and their breakdown products, such as isothiocyanates (ITC) and allyl-isothiocyanate (AITC) subsidize their natural antimicrobial activity. ITC and particularly purified AITC have been widely studied as antimicrobials and have a broad antimicrobial range inhibiting Gram-positive and Gram-negative bacteria, yeasts, and molds [17]. The seed extract of Mustard species (*Brassica* family) was identified to possess coagulation activity against synthetic clay and turbid pond water that can act as a potential natural water treatment agent [13,18,19,20] and can also be a good compliment to Moringa coagulant protein. The antimicrobial activity of Mustard leaf extract is not yet explored.

The current study aimed to investigate the antibacterial effect of Moringa and Mustard leaf extracts and antimicrobial-coagulant protein against thirteen different clinical pathogens isolated from patients samples in India. The Moringa and Mustard leaf were collected in Tamilnadu, Southern India. The coagulant protein was separated by spin column chromatography, and based on LCMS, the protein sequences were identified; the effect of crude extract and coagulant protein on bacterial cell aggregation study was performed by microscopic observation and growth inhibition assay, and the mode of action of the antimicrobial peptide against Gram-negative pathogens was determined by turbidity measurement in a spectrophotometer. Furthermore, the mode of action was determined by applying the simulation docking model.

## 2. Results

The antibacterial activity of Moringa and Mustard crude leaf extract and coagulant protein was analyzed against thirteen clinical pathogenic strains.

### 2.1. Coagulation Activity and Protein Determination

The coagulation activity of *Moringa oleifera* leaf extract (MOS), *Brassica juncea* leaf extract (BJS), *Moringa purified* coagulant protein (MCP), and *Brassica juncea* purified protein (BJP) was measured using kaolin clay solution and clay solution alone used as a control (Figure 1a). The protein concentration was found to be higher in crude extract from MOS than in BJS. The coagulant protein was separated from the crude leaf extracts of Mus and MO by size-excluding spin column chromatography (SESCC), and the eluted protein was concentrated and showed similar coagulation activity as that of crude extract. The molecular mass of coagulant protein was determined by SDS-PAGE (Figure 1b). The coagulant protein separated after SESCC of both MO and Mus showed a protein band of less than ≤3 KDa and an additional band with a molecular mass of around 9 kDa observed in Mus leaf. Based on the Lowry method, the crude protein of MOS (7.0 mg/mL) and BJS (CE) (5.0 mg/mL); likewise, the <3 KDa Purified Protein concentration of MCP (3.6 mg/mL) and Mus CP (2.2 mg/mL).

### 2.2. Antibacterial Activity

#### 2.2.1. Cell Aggregation

Both Gram-positive and Gram-negative clinical isolates were tested for cell aggregation by MOS, BJS extract, MCP, and BJP. All the thirteen bacterial strains were isolated from the patient sample and identified at Ramachandra University, India. Among thirteen different strains tested, MO crude extract (CE) and purified protein (coagulant protein) (CP) showed cell aggregation of ten bacterial strains, whereas Mus CE and CP showed the cell aggregation of eight bacterial isolates. The results from the aggregation experiments are shown in Figure 2.

Both CEs were effective after 2 h of incubation against S. aureus. MCP showed larger flocs of bacterial cells of *E. faecalis*, *P. mirabilis*, and *S. typhimurium* compared to that of MOS. None of the MOS extracts triggered the aggregation of *S. paratyphi* A, K. pneumonie, and S. dysentriae cells. On the other hand, BJS and BJP were not effective against *S. typhi*, *S. paratyphi* A, *S. typhimurium*, *S. marcescens,* and *S. dysentriae*. Likewise, BJP was more effective against *S. flexneri* than BJS extract. In comparison, CE and CP of Mus leaf were effective against *K. pneumoniae*, whereas it was not affected by MOS extracts. Based on the transmission electron microscope, the effect of the antimicrobial peptide on the bacterial cell wall was mainly determined against indicators Gram-positive and negative pathogens such as *E. coli* (ATCC 25922) and *S. aureus* (ATCC 19095) (Figure 2b).

#### 2.2.2. Determination of Minimum Inhibitory Concentration (MIC)

The growth kinetic experiments were performed on all thirteen clinical isolates with CE and CP from Moringa leaf and CP from Mustard leaf (Table 1). Based on the growth inhibitory activity, the minimum inhibitory concentration (MIC) was determined.

The MOS showed growth inhibition up to 6 h against *E. coli*, *E. faecalis*, *S. typhi*, *P. mirabilis*, and *S. aureus*. The concentration of CE ranging between 0.008 and 0.02 mg/mL is required for the complete inhibition of growth. Among the isolates tested, *S.aureus* was affected by 0.02 mg/mL, whereas *P. mirabilis* was effective at 0.008 mg/mL concentration. *S. flexneri*, *S. dysentriae*, *S. paratyphi* A, K. pneumoniae, and Enterobacter sp. showed growth inhibition up to 3 h; thereafter, it started to grow slowly. On the other hand, the optical density was lower as compared to that of the control. The reduced growth rate mentioned in Table 1 refers to inhibited growth up to 3 h (Figure 3).

The growth inhibition assay was performed against five different clinical isolates with MCP. The isolates were selected based on the cell aggregation and effect against MOS. All the strains tested showed a reduced growth rate at the concentration of 0.04 mg/mL (Table 1). Likewise, the BJS extract shows the growth inhibitory activity against all the tested strains with MIC at 0.051 mg/mL.

### 2.3. Identification of Bioactive Protein (Small Molecular Weight–Peptides) by Mass Spectrometry

The peptides that were identified in the low molecular weight peptide profiling and napin profiling are displayed in Appendix A. In total, 2174 peptide in MOP (Appendix A) and BJP 1119 peptide (Appendix A) were notorious and were eluted and tested in vitro for antimicrobial activity; the peptide sequences were screened using an in silico platform [10], peptide profiling was developed in QS 3.0 software (Applied Biosystems, Seoul, Korea) to predict potential antimicrobial peptides (AMPs). Although many potential antimicrobial peptides were identified, the peptides KAVKQQIQQQGQQQGKQQMVSR, FFFLLTN, PPLLQQCCNEL, SGGGPS, and GPQQRPPLLQQCCNELHQEEP (Appendix A) were the most abundant. These peptides showed the strongest antimicrobial activity (1.5 mg/mL) in the ≤ 3 KDa fraction of the plant extract (Appendix A, and Appendix A), and they were not much different in their inhibitory abilities (*p* > 0.05) when compared with commercial napin (Sigma 14-837 MSDS) (Figure 2a,b and Appendix A). Napin showed stronger inhibitory activity against food-borne pathogens (Figure 2a,b). The development of AM mechanisms in peptides against food-borne pathogens remains an important global public health.

### 2.4. Molecular Interaction of AM with Pathogens (Mode of Action)

#### SWISS-MODEL: Homology Modeling of Protein Structures

Molecular dynamics were considered to be an efficient tool to form a simulation model to find the linkage between plants-based bioactive (napin Peptides—small molecular size proteins, which is produced as a secondary metabolite) compounds and their respective targets, enabling interactions at the specific period. The 3-dimensional quantitative modeling (3D-QSAR), to understand the relationship of specific amino acids of progress in 3D quantitative structural activity relationship modeling (3D-QSAR) has reopened a method to compare the plant’s secondary metabolites and their targeted molecular function. The 3-dimensional structure of napin peptide analog 5 (Figure 4) made a sequence based on similar proline and glutamine-rich peptide motifs.

According to the in silico molecular modeling docking analysis, the first loop region of napin was formed by three different amino acid such as Isoleucine (Ile), Serine (Ser), and Proline (Pro) between the 18th and 20th positions, whereas the secondary loop is based on Glycine (Gly), Glutamine (Gln), Valine (Val), and Glycine (Gly) between the 39th and 41st positions (Figure 5) (Table 2). The secondary structure of napin peptide, the electrostatic potential of the protein with various ligands. The electrostatic potential was calculated using the Poisson–Boltzmann equation-based algorithm (Figure 4a–c). The blue color represents the positive domain’s electrostatic potential of 3.0 kcal/mole (total charge of +5). However, the electrostatic potential of the less negative domains was represented in red and documented in a three-dimensional structure. The ambiguous comportment of a positively charged protein networking indicated an electrostatic interaction between the negative domains and positively charged ions.

Based on the Cluspro model scores, the top 5 model scores are taken for the calculation of interaction between proteins (Table 2). Using Surface Racer, the molecular surface area of each docked model is calculated. Based on the surface area and lowest energy provided by Cluspro for each cluster, the final model was selected.

The hydrogen and hydrophobic interaction plot of the protein is represented in the Ligplot+ figure, while the residue and type of interactions involved in the interface are reported in Table 3, which represent the binding strength and stability of the docked complex of napin with the other three proteins.

### 2.5. Antimicrobial Resistance

The antimicrobial activity of the plant extract was compared with the standard antibiotics, as shown in Table 4. Gentamicin, ampicillin, and vancomycin were highly effective against the tested bacterial strains, while gentamicin was found to have inhibitory activity, similar to MOS extract (Table 5). Bacteria were least sensitive toward Novobiocin and Clindamycin; however, gentamicin displayed a variable trend. In general, all the tested antibiotics showed a similar zone of inhibition except for ampicillin and vancomycin. In terms of comparison with a natural plant-based antimicrobial agent, it was effective against all the microorganisms tested in this study. Thus, the plant extract could be an attractive alternative for the preservation of food as a natural antimicrobial agent.

## 3. Discussion

These results based on coagulation activity and protein determination were found to be correlating with earlier reports on coagulant protein from MOS and Mus [11,12,20]. On the basis of cell aggregation, the antibacterial effect of both BJS and BJP were effective against *K. pneumonia*, whereas it was not affected by MOS extracts. The possible reason could be that BJS might have more than one leaf storage protein that has coagulation properties that could play a role in the aggregation of bacterial cells, as it is observed that the eluted fraction from the ion exchange matrix revealed two protein bands having a molecular mass of 6.5 and 9 kDa. Aluko [21] reported that the peptide sequence of 9 kDa protein from BJS was similar to napin3 protein, which has sequence homolog to the coagulant protein. Further study is needed to understand the mechanism of action against bacterial strains. Broin et al. [22] reported that the minimal inhibitory concentration (MIC) for *E. coli* was 50 mg/mL up to 150 min. In this study, the MOS showed growth inhibition at a low concentration of 0.008 mg/mL up to 6 h, proving the efficiency of the proteins present in the crude extract. The possible reason could be the extraction method used to get soluble proteins to form the crude extract and/or the variety of the leaf used in this study and the efficiency of coagulant protein.

Based on the growth inhibition assay, when using a higher concentration of MOS, an enhanced growth rate was observed. An example can be seen in Figure 3: the growth is enhanced when a high concentration of CE is used against *S. marcescens*. Similar results were observed with other strains tested in this study. This can be explained that the crude extract could serve as a nutrient source for the microorganisms. This is very well in correlation with earlier reports that stated that when drinking water is treated with MOS and stored for a longer time, it will change the taste and odor due to organic load and disinfection by-products [23,24]. Therefore, it is essential to separate the coagulant protein from the crude extract and test for antibacterial activity and water treatment.

In comparison, the MCP alone is not as effective as that of MOS extract. Earlier reports on MCP showed growth inhibition against *E. coli* [25,26,27]. The reason for the differences in the behavior could be that different bacterial isolates were selected for the antibacterial test of MCP for instance, Silveira [28] and Williams [29] reported using *E.coli* D31 and *Bacillus thuringiencis* strain Bt7 and showed growth inhibition up to six hours. In the present study, the pathogenic organisms tested for antibacterial activity were isolated from patients. The other possible reason could be the variety of *Moringa* leaf used for isolating the coagulant protein. In the present study, the *Moringa* leaf was collected in Southern India, whereas the seeds collected from different places such as Kenya and other geographical regions were reported [30]. Further study is needed to explore the differences in coagulant protein from different localities and varieties of MOS by protein sequence comparison and mechanism of action against different pathogenic strains. 

MusP leaf showed growth inhibition up to 6 h against *S. flexneri* and *S. paratyphi* A. On the other hand, *E. coli*, *E. feacalis*, and *S. dysentriae* showed reduced growth. At 6 h of incubation, there is a reduction of turbidity between control and coagulant protein treated cells. MusP did not affect the other microorganisms tested (Table 1). The growth inhibition assay for BJS was not performed, since the extract has color and interfered with the absorption measurement. When compared with MOS, BJS was not effective against most of the organisms tested, stating that the antimicrobial activity of MOP and MusP (Purified proteins) is more promising.

Interestingly, in certain cases, both the leaf extracts showed cell aggregation but lacked bacterial growth inhibition (Table 2). For instance, MOS showed aggregation of *S. paratyphi B, S. typhimurium,* and *S. marcescens* but showed no inhibition of bacterial growth. Concerning BJS, it showed activity during cell aggregation experimentations but failed to do so during cell inhibition studies in isolates such as *S. parathypi B, K. pneumonia, S. aureus, E. species,* and *P. mirbalis.* These organisms are only partially sensitive to crude extracts. To our knowledge, this is the first report on BJS showing bacterial cell aggregation and reduction in growth rate against clinical isolates. 

The pathogenicity and resistance of the bacteria toward the antibiotics and virulence factor such as K1antigen, enterochelin, type 1 pilli, P-pili, hemolysin, and col V plasmid [30] plays a major role in their characterization [31]. Microbes are evolving to be resistant against antimicrobial agents that are currently active [32]. Hence, new antimicrobial agents such as plant peptides, resins, and other secondary metabolites are likely to be emerging to act as antimicrobial agents against otherwise resistant microbes [24,33,34,35,36,37]. Most of the plant materials act as antibacterial agents. It has been reported that secondary metabolites such as peptides, proteins alkaloids, phenols are known to have antimicrobial properties [24,35,36,37]. A few of the many virulences and antigenic proteins are O flagella and H antigens [38,39] Incompatibility group H (IncH1 and IncH2) antigens of *Salmonella species* on their H-pilus structures [39]. In the case of *Salmonella*, there are over 2000 serotypes, based on their antigenic variation associated with gastroenteritis and enteric (typhoid) fever in humans [40,41]. Although the roles of these pathogenic proteins are highly classified and understood, we speculate that such properties such as antigenicity, virulence plasmid, and environment plays a vital role in determining the bacteriostatic and bacteriocidic activity for different leaf extracts that are being investigated. Such an observation was observed in *S. paratyphi* A and B showed variation in response toward the moringa and mustard protein. Likewise, *S. flexneri* and *S. dysentriae* showed variation in response to protein; it may be that the pathogenesis of *S. flexneri* varies based on invasion plasmid antigens (Ipa A, B, C, D, H), surface presentation antigens (Spa), membrane excretion proteins (Mxi A, B), and virulence proteins (Vir B, G), but when compared with *S. dysenterae*, their pathogenesis may vary based on toxin (endotoxin) production [42,43,44].

Among the diverse classes of antimicrobial-based storage peptides, low molecular weight peptides were specifically found to be rich in Proline (Pro) and were documented with higher potency and specificity. Pro is a non-polar amino acid that poses a specific target on the interfacial region over the membrane bilayer. The storage peptide rich in pro-residue stabilizes the non-polar region and extends the stability of attachment to the bacterial membrane [45].

The current study explores the discrepancies in describing the molecular weight and sequence profiling of antimicrobial peptides (small molecular weight proteins) from MOS and BJS extract through laboratory as well as in silico analyses. The results specify the existence of monomer and dimer forms, which exhibit the same antimicrobial activity. This observation is constant with previous reports [46,47]. It is worth stating the presence of a band between 20 and 24 kDa, while high-intensity bands reported around <10 kDa were detected [46,48]. The oligomers were linked with S–S bonds and further future research was initiated toward understanding the oligomerization of *napin* protein through protein–protein docking and further confirmed using LC-ESI-TOF-MS/MS (Appendix A). There were inadequate structural interaction studies relating to the functional activity of MOP and MusP low molecular weight peptide (<3 kDa) [48]. In 2018, Hald et al. [49] proposed a 70% structurally similar protein based on the peptide sequence of *napin* determined by LC-TOF-MS, but modeling and the mechanism determination were lacking. The current work intends to propose a simple and most reliable homology modeling; we are propositioning here the structure of MOP and MusP sequence with 99% similarity (Appendix A and Figure 1 and Figure 2) by Pinto et al. [50]. The capability of this protein to form oligomers was examined through a protein–protein docking study. 

To find out whether the coagulant protein triggers any resistance development of the microorganisms, two bacterial strains were selected, namely *E. coli* and *S. paratyphi* A, where the former showed good growth inhibition and latter one has no effect. The organisms were incubated with coagulant protein and isolated repeatedly three times. The isolated colonies were growing as equal to that of control cells, implying that no resistance developed. *Moringa* coagulant protein has been reported as a complement to the chemicals in the drinking water treatment process. The present study reveals that the antimicrobial property of MOS and BJS extracts provides a good substitute to treat water, thereby reducing the risk of infection caused by water contamination.

## 4. Materials and Methods

### 4.1. Materials and Chemicals

The following materials, chemicals, solvents, and drugs were used: Paper Disc (Advantec, India), Disk Diffusion Zone (Inhibitor) measuring ruler (Himedia, Chennai, India), Ethanol, Methanol (Fisher Scientific, Chennai, India), 96% Ethanol (Fisher Scientific, Gangnam, Seoul, South Korea), Mueller–Hinton Agar (Himedia, Chennai, India), Nutrient Broth (Himedia, Chennai, India), Mannitol salt agar (Himedia, Chennai, India), Triptic soy agar Phosphate buffer Saline (Fisher Scientific, Chennai, India), Antibiotics (ampicillin, vancomycin, penicillin, gentamicin, tetracycline, kanamycin, clindamycin, erythromycin, and novobiocin (Himedia, Chennai, India).

### 4.2. Isolation of Pathogenic Microorganisms from Patient’s Sample

The clinical samples such as urine, blood, stool, throat swab, and sputum were rolled on the culture plates (Nutrient Agar, Blood Agar, MacConkey Agar, and Chocolate Agar) for the isolation of Pathogenic Microbes, which were incubated overnight and examined for the growth. If there was no growth in 24 h, the plates were incubated further. Organisms were identified based on biochemical reactions (Coagulase, Indole, Citrate, Urease, Mannitol Motility, Triple Sugar Iron Agar, and Phenyl Alanine Deaminase) and colony characteristics. 

### 4.3. Microbial Strains Applied in the Study

Clinical isolates of thirteen bacterial strains—Escherichia coli (*E.coli*), Salmonella paratyphi B (*S. paratyphi* B), Enterobacter species (*E. species*), Shigella flexneri (*S. flexneri*), Salmonella typhimurium (*S. typhimurium*), Salmonella typhi (*S. typhi*), Salmonella paratyphi A (*S. paratyphi* A), Klebsiella pneumonia (*K. pneumonia*), Staphylococcus aureus (*S. aureus*), Serratia marcescens (*S. marcescens*), Shigella dysentriae (*S. dysentriae*), Proteus mirbalis (*P. mirbalis*), and Enterococcus faecalis (*E. faecalis*)—were obtained from Department of Microbiology, Ramachandra Hospital in Chennai, India. Stock culture was maintained in nutrient agar media at 4 °C.

### 4.4. Extraction of Crude Protein (Active Compounds) from Plant Leaf

*Moringa oleifera* (MOS) and Mustard *Brassica juncea* (BJS) leaf were purchased from local shops in Southern India. The preparation of crude extracts from *Moringa* and Mustard leafs were performed as described earlier [19]. Dried MOS and BJS were grounded into fine powder by using mortar and pestle. To remove oil from the fine powder, 95% ethanol was added, and the supernatant was separated by centrifugation at 3000 rpm for 10 min. The pellet was allowed to air dry, and 5% aqueous extract was prepared using sterile distilled water. The obtained soluble fraction is referred to as crude extract (CE). 

### 4.5. Bioactive Protein—Functional Activity

#### 4.5.1. Coagulation Activity

The crude extracts and the coagulant proteins were tested for coagulation activity using synthetic clay solution as described earlier [11]. Clay solution (1%) was prepared using kaolin clay and mixed with protein to get a final volume of 1 mL and initial optical density at 500 nm and after 60 min was measured. The percentage of coagulation activity was calculated using the formula: (Coagulation activity % = [(Initial absorbance − final absorbance) / initial absorbance] × 100).

#### 4.5.2. In Vitro Assay for Antimicrobial Activity

##### Disc Diffusion Kirby Bauer Method

The antagonistic effect of *Moringa oleifera* crude protein (MOS) 0.016, 0.02, 0.03, 0.04, and 0.08 mg/mL; *Brassica juncea* crude protein (BJS) 0.017, 0.02, and 0.05 mg/mL; *Moringa oleifera* antimicrobial-coagulant Protein (MCP) 0.006, 0.012, 0.015, and 0.03 mg/mL, and *Brassica juncea* antimicrobial-coagulant Protein (BJP) 0.001, 0.015, 0.02, and 0.05 mg/mL was identified using selected enteropathogens acquired from Department Food Science and Biotechnology, Anna University, India. The clinically isolated pathogenic strains were applied to a antimicrobial activity measuring disk that was 8 mm in diameter. The spread plate method was used to inoculate each microorganism on nutrient agar plates. The plates were incubated at 37 °C for 24 h, and the diameter of the zone of inhibition was measured and compared to evaluate the antibacterial potential of the different extracts [51].

##### Growth Inhibitory Activity—Determination of Minimum Inhibitory Activity (MIC) and Minimum Bactericidal Activity (MBC)

The clinical isolates were grown in nutrient broth (NB) and incubated at 37 °C in a shaking incubator for overnight and diluted to an initial optical density (OD) of 0.1 and 0.3 at 600 nm for growth inhibition assay and cell aggregation test, respectively. The concentrations of the protein that were used for cell aggregation tests were MOS 0.016, 0.02, 0.03, 0.04, and 0.08 mg/mL; BJS 0.017, 0.02, and 0.05 mg/mL; MCP 0.006, 0.012, 0.015, and 0.03 mg/mL and Mus CP 0.001, 0.015, 0.02, and 0.05 mg/mL. Furthermore, the CE and CP were added to the culture suspension with an initial OD of around 0.3 and incubated at 37 °C for 4 h. The cell aggregation was observed at each hour under the phase-contrast microscope (Nikon Eclipse 80i, Japan), and the images were captured using a microscopic digital camera (EM- 200F, USB-2.0, CCD Chip). The growth kinetics studies were conducted by adding different concentrations of protein to pre-diluted overnight cultures (0.1 OD) to a final volume of 1 mL in a sterile cuvette and incubated at 37 °C for 4–6 h with continuous shaking. The OD at 600 nm was measured every 60 min by spectrophotometer (Eppendorf) and values were recorded [51]. 

#### 4.5.3. Electron Microscopy Analysis

Bacterial cells were set at 30 ± 2 °C for 1 h with 2% glutaraldehyde and paraformaldehyde. After many washes with 0.1 M cacodylate buffer, cells were dehydrated with ethanol [51]. Then, bacterial cells were injected with increasingly concentrated Eponate 812 and then polymerized at 60 ± 2 °C for 2 days. Sectioned with ultra-microtome and stained with uranyl acetate using field-emission transmission electron microscope (FETEM) (JEM-2100F, JEOL) KBSI, Chuncheon, Gangwon-do, Korea.

### 4.6. Purification of Coagulant and Antimicrobial Protein

#### 4.6.1. Purification of Protein Based on Column Chromatography 

Based on size-exclusion spin column chromatography, a < 30 KDa spin column with < 3 KDa filtrate was subsequently applied to separate the low molecular weight (< 3 KDa) antimicrobial peptides (AMPs).

#### 4.6.2. Tris-Tricine SDS-PAGE Electrophoresis and Gel-Elution

Furthermore, Tris-Tricine SDS-PAGE (polyacrylamide gel electrophoresis) [21] was applied to separate the <10 KDa peptide band, and based on gel elution, a <3 KDa peptide band was eluted and proceeded for LC-ESI-TOF-MS/MS (mass spectrometry liquid chromatography-electrospray ionization-quantitative time-of-flight tandem mass spectrometry). 

#### 4.6.3. Quantification of Protein Concentration

The protein content of MOS and BJS crude extracts and coagulant protein was estimated by Lowry’s method [19]. The purity of the coagulant protein was analyzed by Tris-Tricine SDS-PAGE gel analysis according to Laemmli [20]. The gels were stained with Coomassie brilliant blue to visualize the protein. 

#### 4.6.4. Identification of Purified Peptides by Mass Spectrometry

Sequential identification of peptides by LC-ESI-TOF-MS/MS (mass spectrometry liquid chromatography-electrospray ionization-quantitative time-of-flight tandem mass spectrometry) were analyzed at the National Instrumentation Center for Environmental Management of Seoul National University in Seoul, Korea, according to an earlier method by Chelliah et al. [51]. The peptides were inspected using QS 3.0 software (Applied Biosystems, Seoul, Korea). The ranges of proteins were identified based on 300–3000 *m*/*z* values. 

### 4.7. SWISS-MODEL: Homology Modeling of Protein Structures

The protein models were spawned by SWISS-MODEL, licensed under the Creative Commons Attribution-Share Alike 4.0 (CC BY-SA 4.0) International License, based on the sequence generated by liquid chromatography-mass spectrometry (LCMS). SWISS-MODEL generates theoretical models by automated homology modeling techniques developed by the Computational Structural Biology Group at the Swiss Institute of Bioinformatics (SIB) at the Biozentrum, University of Basel, Basel, Switzerland.

#### In Silico—Molecular Interaction Analysis and Docking for Antimicrobial Peptides in MOS

In silico molecular docking was performed to assess the docking ability of napin with Lipid II, 4PLB, and LipoXc. The 3D structure of *napin*, Lipid II, 4PLB, and LipoXc is downloaded from RCSB PDB. Proteins were prepared before docking and provided to Cluspro (v 2.0) [52]. The Cluspro web server is a widely used tool for protein–protein docking. It works on the fast Fourier transform correlation approach where simple scoring functions evaluate docking confirmation. Cluspro performs multi-stage protocol: rigid-body docking, energy-based filtering, evaluating structures based on clustering properties, and finally returning a small number of structures based on minimized energy. The server returns models based on energy and cluster size, among which one of the returned models was selected based on the lowest energy and size of the cluster.

Binding energy surface properties are calculated, which in turn give the protein interface probability and the interaction site of two proteins. For surface properties calculation, Surface Racer 5.0 is used, which calculates the exact Accessible Surface Area (ASA), Molecular Surface Area (MSA), and cavities to the inner protein inaccessible to solvent from the outside. The output includes the surface area of the docked protein models for each residue in addition to those of individual atoms [53].

The top five ranked clusters are taken from Cluspro and run in the surface racer using van der Waal’s radii as 2, and the radius for all the models was taken as 1.4 for surface area calculation. From the surface racer output, MSA was considered, and the best matching model is calculated. The difference between the MSA of the docked proteins model from the sum of the surface area of individual proteins and the larger surface area model is taken for further visualization of interactions.

Furthermore, the interacting bonds within two docked protein complex molecular interactions within the docked protein complex were observed. Ligplot (v 2.2) [53] was used under an academic user license, and the DIMPLOT program of LigPlot + was executed to obtain the interactions across a selected protein–protein docked complex. The hydrophobic interactions and hydrogen bonds between the two docked proteins are represented in the figure where interacting residues are reported in Table 3. A computer-based molecular docking method was performed to identify the main compounds responsible for the antibacterial activity of ethanol extracts of different samples based on their possible binding mode and theoretical affinity toward bacterial topoisomerases (4PLB) and bacterial cell wall lipoprotein complex. The Cluster 2.0 web server web program for ligand–protein interaction is based on the fast Fourier transform correlation method that calculates docking validation by simple scoring functions [53,54].

### 4.8. Antimicrobial Peptides (AMPs) with Limited Resistance

Our aim was to verify the inheritable nature of the evolved resistance from bacteria encountering AMPs frequently in their natural environments and evolving mechanisms to resist their action. Intrinsic resistance to AMPs can occur via passive or inducible mechanisms; we reconditioned the bacterial strain that showed good aggregation property, and one bacterial strain that did not show an aggregation property was selected to test the resistance development against coagulant protein. *E. coli* cells were incubated with MOS, BJS, MCP, and BJP for 7 h at 37 °C. The cells were isolated from the culture and incubated again with protein for a growth inhibition test. This cycle was repeated 3 times to check the possibility of bacterial resistance development against coagulant protein. The limits of tolerance were confirmed as established in previous assays.

### 4.9. Statistical Analysis 

Then, the collected data were analyzed using ANOVA and the SD method, which were performed to determine the specific consequence in variances of MOL and BJS extract on measurable constraints (*p* < 0.05) according to Okada [55]. All tests were repeated in triplicates.

## 5. Conclusions

The effect of crude extract and coagulant protein from *Moringa* and Mustard leaf were investigated against thirteen pathogenic microorganisms. Crude extract and coagulant protein from Mustard leaf protein has antibacterial activity when measured by cell aggregation and growth inhibition studies. *K. pneumoniae* was sensitive to crude extract and coagulant protein from Mustard leaf in terms of cell aggregation. Furthermore, it is shown that even low concentrations of MOS and BJS extracts could effectively have antibacterial activity. In addition, based on spin column chromatography and LC-ESI-TOF-MS/MS, the bioactive (antimicrobial and coagulant activity) low molecular weight protein (peptide) was identified as *napin* (storage protein) found both in *Moringa* and mustard leaf. Due to the virulence and antigenic properties, the stain differentiation could play a major role in response toward the *Moringa* and Mustard leaf extracts. The organisms did not develop resistance against the coagulant protein. Furthermore, the elucidation of the mechanisms behind such aggregation and inhibition activities would widen the understanding of such mechanisms and thus identify similar coagulant molecules that can be effectively used in the water treatment process. However, there is still a need for more in vitro and in vivo studies to verify the positive roles for the human health of the isolates acquired in this research.

## Figures and Tables

**Figure 1 molecules-26-02080-f001:**
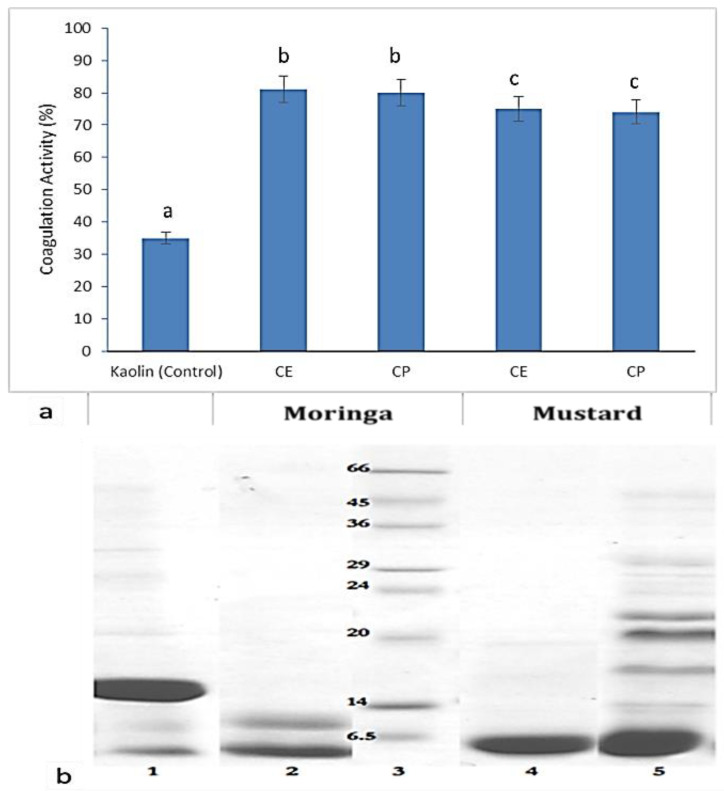
(**a**) Coagulation activity of Moringa oleifera leaf extract (MOS) and Brassica juncea leaf extract (BJS) and coagulant protein (CP) after 90 min of sedimentation. The crude extracts (CEs) were not diluted, whereas CPs were five times diluted. Kaolin clay without the addition of extract served as a control. Values are expressed as the mean ± standard deviation (n = 3). Different superscripts (a–c) represent significantly different values (*p* < 0.05). (**b**) SDS-PAGE of BJS and MOS extracts. Lane 1: *Moringa oleifera* leaf extract (MOS), 2: *Moringa purified* coagulant protein (MCP), 3: Marker sigma-aldrich 6.5–66 kDa, 4: *Brassica juncea* purified coagulant protein (BJP), 5: *Brassica juncea* leaf extract (BJS).

**Figure 2 molecules-26-02080-f002:**
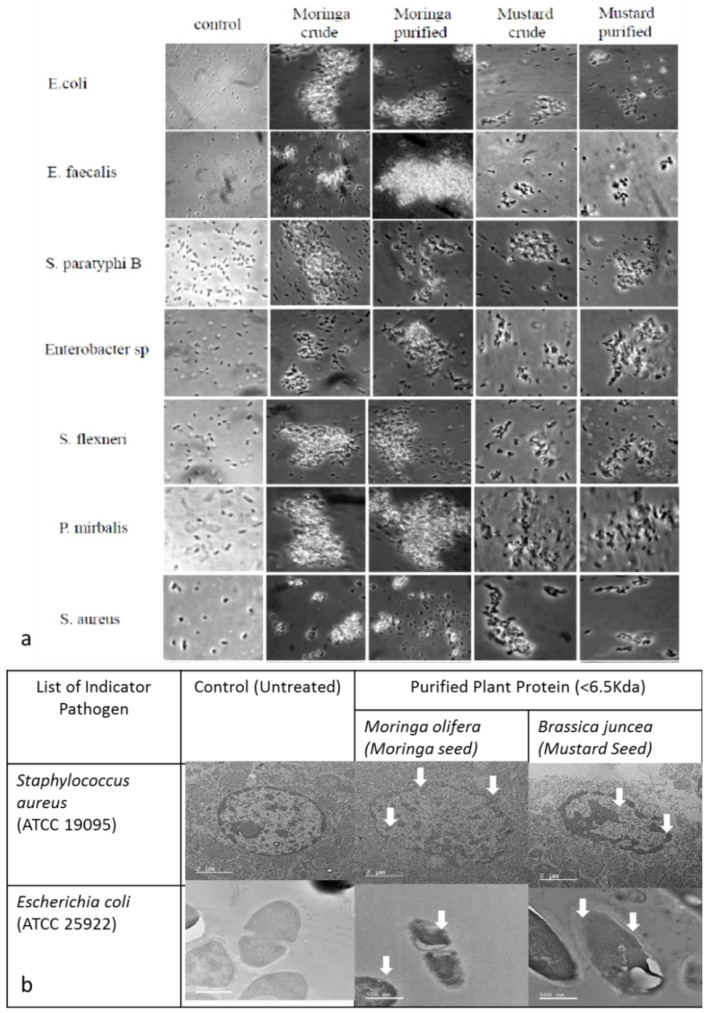
(**a**) Comparative analysis for aggregation experiments of MOS, BJS (extract), MCP, and BJP (protein) toward thirteen different clinical pathogens. (**b**) Transmission electron microscopic imaging indicates the antimicrobial activity of purified plant protein (<6.5 KDa) against *E. coli* (ATCC 25922) and S. aureus (ATCC 19095).

**Figure 3 molecules-26-02080-f003:**
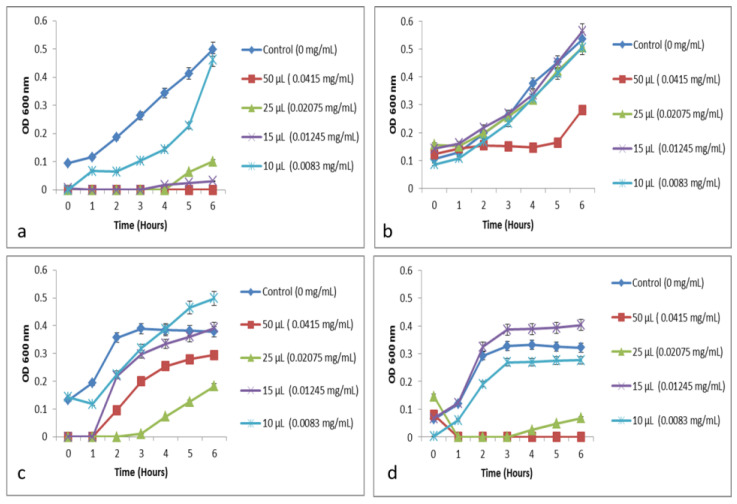
Growth inhibition of pathogens based on the different concentrations, further minimum inhibitory concentration was determined in Table 1. (**a**) MOS extract growth inhibitory activity of S. marcescens, (**b**) BJS growth inhibitory activity of S. marcescens, (**c**) MOS extract growth inhibitory activity of S. flexneri, (**d**) BJS growth inhibitory activity of S. flexneri. Values are expressed as the mean ± standard deviation (n = 3).

**Figure 4 molecules-26-02080-f004:**
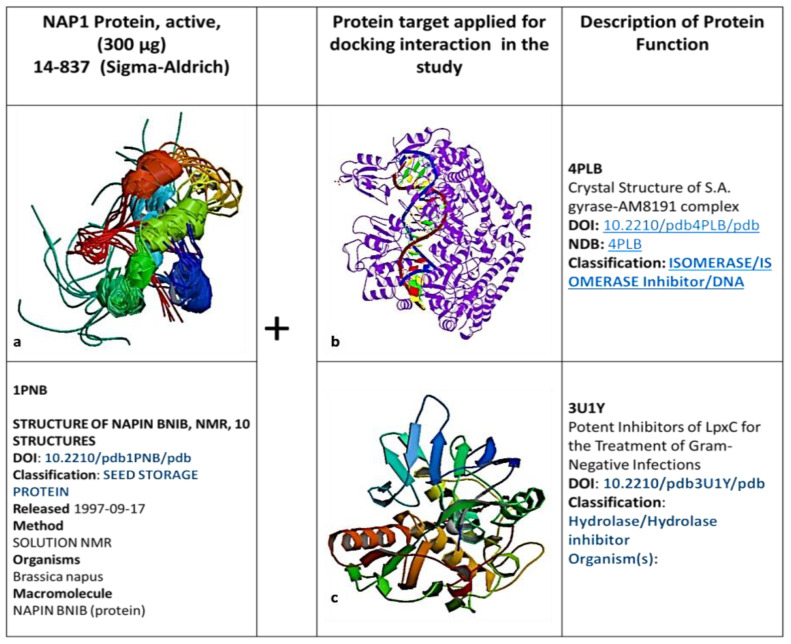
Regarding molecular docking ligand and protein structure were determined from the respective data bank. (**a**) Three-dimensional (3D) molecular structure of the structure of *napin* was determined from (www.rcsb.org/structure/1PNB). (**b**) Two-dimensional (2D) molecular structure of S.A. gyrase–AM8191 complex was determined from (www.rcsb.org/structure/4PLB). (**c**) Crystal structure of potent inhibitors of LpxC (treatment of Gram-Negative infection dimensional (3D) Molecular structure was determined from (www.rcsb.org/structure/3U1Y).

**Figure 5 molecules-26-02080-f005:**
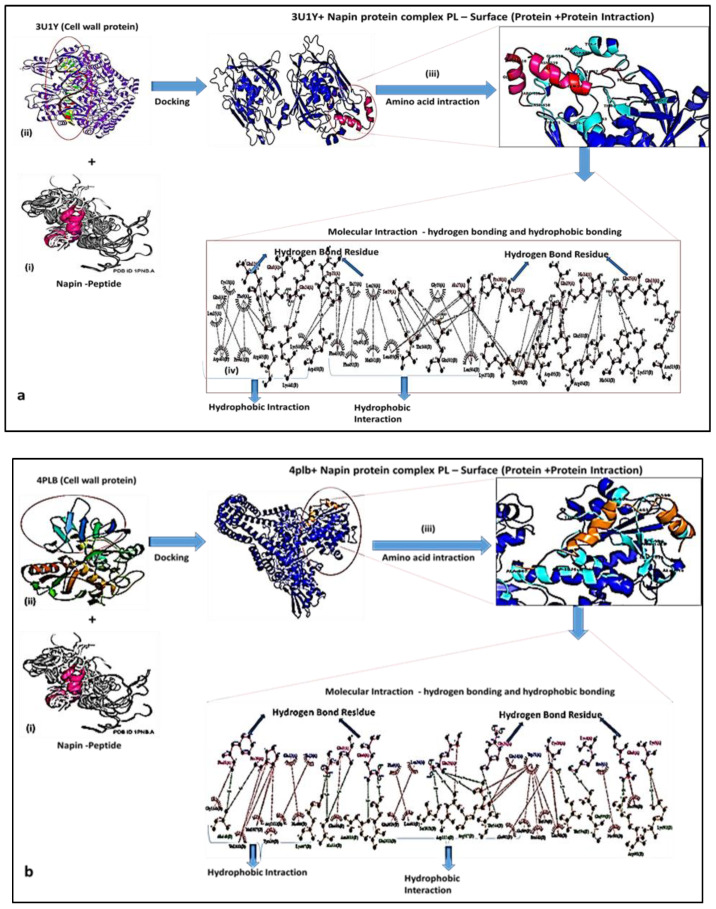
(**a**) (i) The interaction analysis between *napin* (pink) and 3U1Y (Blue) by molecular docking. (ii) The interaction model of complex *napin* and 3U1Y—the 3D structure of protein complex. (iii) Interfaces and key residues analysis. (iv) Ligplot representation of showing hydrogen and hydrophobic interactions between Pediocin and LipoXc (cell-wall protein) complex. (**b**) (i) The interaction analysis between *napin* (pink) and 4plb (Blue) by molecular docking. (ii) The interaction model of complex *napin* and 4plb—the 3D structure of protein complex. (iii) Interfaces and key residues analysis. (iv) Ligplot representation of showing hydrogen and hydrophobic interactions between Pediocin and 4plb (Topoisomerase peptide) complex.

**Table 1 molecules-26-02080-t001:** The growth inhibitory activity of Moringa olifera crude extract (MOS), Brassica juncea extract (BJS), Moringa olifera purified protein (MCP), and Brassica juncea purified Protein (MusP) against different clinical pathogens. Antimicrobial activity of plant crude extract and purified coagulant proteins.

SNo	List of Microorganism	Minimum Inhibitory Concentration (MIC)			
		*Moringa**olifera* Leaf		*Brassica juncea* Leaf	
		Crude Protein (MOS) (mg/mL)	Coagulant Protein (MCP) (mg/mL)	Crude Protein (BJS) (mg/mL)	Coagulant Protein (BJP) (mg/mL)
1	*E. coli*	0.01	0.02	0.05	0.09
2	*E. faecalis*	0.01	0.02	0.02	0.04
3	*S. paratyphi B*	0.04	0.06	0.05	0.09
4	*E. species*	0.01	0.02	0.05	0.08
5	*S. flexneri*	0.02	0.04	0.04	0.07
6	*P. mirabilis*	0.008	0.01	0.05	0.07
7	*S. aureus*	0.01	0.02	0.05	0.09
8	*S. typhimurium*	0.04	0.06	0.03	0.06
9	*S. typhi*	0.02	0.04	0.05	0.08
10	*S. marcescens*	0.01	0.02	0.05	0.08
11	*K. pneumoniae*	0.01	0.02	0.05	0.09
12	*S. paratyphi A*	0.02	0.04	0.02	0.04
13	*S. dysenteriae*	0.01	0.02	0.05	0.08

**Table 2 molecules-26-02080-t002:** List of interacting residues between two docked proteins. Interaction type is mentioned as hydrophobic interactions and hydrogen bond interaction between napin with 3U1Y and 4PLB.

**Type Of Interactions**	**Napin**	**3u1y**	**Total**	**NPN-3U1Y**	**Total-MSA**	**Weighted Score**	**Representative**
Napin_3u1y	2768.19	25424.35	28192.54	25088.38	3104.16	−786	Lowest Energy
	**Napin**	**4plb**	**Total**	**NPN-4PLB**	**Total-MSA**	**Weighted Score**	**Representative**
Napin_4plb	2768.19	54046.17	56814.36	53812.11	3002.25	−912.9	Lowest Energy

Strongest Binding: Napin–Lipid2; Weak Binding: Napin–3u1y.

**Table 3 molecules-26-02080-t003:** Molecular instructions of *napin* with compounds with bacterial structural and function proteins (based on LC-ESI-TOF-MS/MS analysis among the 2174 peptide in MOS and 1119 peptide in BJS; the *napin* peptide sequence similarity was found to be higher and stronger with molecular docking interaction with the cell wall of the Gram-negative bacterial protein (LpxC) and topoisomerase peptide (4PLP)).

Type of Interactions	Hydrogen		Hydrophobic	
**Chain ID**	**Chain A (Napin)**	**Chain B (3U1Y)**	**Chain A (Napin)**	**Chain B (3U1Y)**
**3U1Y**	GLU 12, GLU 8 GLN 24, TRP 21 SER 29, ALA 27 PRO 30, ARG 23 GLN 19, HIS 14 GLN 25, GLN 10	ARG 465, LYS 441, ASP 458, THR 360, GLN 502, LYS 371, TYR 498, ASP 495, ARG 494, GLU 518, HIS 563, LYS 537, ASN 519	EU 15, GLN 6 CYS 18, PHE 9 ILE 22, LEU 26 GLY 28	ASP 460,PRO 462, LYS 560, PHE 459, PHE 4922, GLY 491, MET 361, LEU 499, LEU 504
**Chain ID**	**Chain A (Napin)**	**Chain B (4PLB)**	**Chain A (Napin)**	**Chain B (4PLB)**
**4PLB**	PHE 31, PRO 30, GLN 1, GLN 6, GLN 25, GLN 20, CYS 18, LYS 4, GLU 8, CYS 5	ALA 640, LYS 607, ALA 534, ASN 1010, GLU 1011, SER 1021, ASP 1024, ARG 517, THR 544, THR 594, GLU 599, ARG 601, LYS 502	GLU 12, ILE 22, PHE 9, LEU 26, GLN 24, TRP 21, PRO 2	GLY 1341, VAL 1031, MET 1027, ARG 1012, TYR 639, HIS 600, GLN 505, GLU 1020, LEU 603, ALA 602, ALA 509, PRO 542, ILE 539, GLN 541, HIS 501, PRO 598

**Table 4 molecules-26-02080-t004:** Comparative analysis of antibiotics toward thirteen different clinical pathogens (analysis on the antimicrobial resistance of clinical pathogens (isolated from dysentery patients), which was applied in the study).

SNo	List of Pathogens	Nucleic Acid Targeting Antibiotic (A)	Cell Wall Targeting Antibiotic (B)			Protein Synthesis Inhibiting Targeting Antibiotic (C)
		Novobiocin (mm)	Clindamycin (mm)	Gentamicin (mm)	Ampicillin (mm)	Vancomycin (mm)
1	*E. coli*	12.00 ± 0.05 ^c^	14.00 ± 0.10 ^b^	16.00 ± 0.05 ^b^	12.00 ± 0.07 ^c^	13.00 ± 0.05 ^c^
2	*E. faecalis*	14.00 ± 0.03 ^a^	-	17.00 ± 0.02 ^b^	12.00 ± 0.01 ^c^	12.00 ± 0.03 ^c^
3	*S. paratyphi B*	-	-	-	16.00 ± 0.01 ^b^	10.00 ± 0.07 ^c^
4	*E. species*	-	-	13.00 ± 0.05 ^c^	14.00 ± 0.03 ^b^	15.00 ± 0.03 ^b^
5	*S. flexneri*	-	-	15.00 ± 0.03 ^b^	15.00 ± 0.04 ^a^	17.00 ± 0.03 ^b^
6	*P. mirabilis*		-	17.00 ± 0.03 ^b^	16.00 ± 0.03 ^b^	14.00 ± 0.11 ^b^
7	*S. aureus*	-	-		14.00 ± 0.08 ^b^	15.00 ± 0.02 ^b^
8	*S. typhimurium*	-	-		15.00 ± 0.03 ^b^	-
9	*S. typhi*	16.00 ± 0.05 ^a^	17.00 ± 0.02 ^a^	12.00 ± 0.03 ^c^	14.00 ± 0.09 ^b^	12.00 ± 0.03 ^c^
10	*S. marcescens*	12.00 ± 0.12 ^c^	15.00 ± 0.06 ^a^	15.00 ± 0.03 ^b^	12.00 ± 0.04 ^c^	10.00 ± 0.02 ^c^
11	*K. pneumoniae*	-	-	-	12.00 ± 0.03 ^c^	13.00 ± 0.05 ^c^
12	*S. paratyphi A*	-	-	-	14.00 ± 0.05 ^b^	12.00 ± 0.01 ^c^
13	*S. dysenteriae*	-	14.00 ± 0.04 ^b^	14.00 ± 0.03 ^c^	10.00 ± 0.02 ^c^	10.00 ± 0.06 ^c^

- not active, ^a^ more sensitive, ^b^ moderate sensitive, ^c^ less sensitive, Media—tryptic soy agar.

**Table 5 molecules-26-02080-t005:** Overall view on the combined effect of coagulation (flocculation of artificial clay–kaolin) and antimicrobial activity (growth inhibition) of *Moringa olifera* (*Moringa*) and *Brassica juncea* (Mustard) leaf extracts toward thirteen different clinical pathogens.

Sample SNo:	List of Pathogens	*Moringa olifera* *(MOS)*		*Brassica juncea* *(BJS)*	
		Aggregation	Growth Inhibition	Aggregation	Growth Inhibition
1	*E. coli*	+	+	+	+
2	*E. faecalis*	+	+	+	+
3	*S. paratyphi B*	+	-	+	-
4	*E. species*	+	+	+	-
5	*S. flexneri*	+	+	+	+
6	*P. mirabilis*	+	+	+	-
7	*S. aureus*	+	+	+	-
8	*S. typhimurium*	+	-	-	-
9	*S. typhi*	+	+	-	-
10	*S. marcescens*	+	-	-	-
11	*K. pneumoniae*	-	+	+	-
12	*S. paratyphi A*	-	+	-	+
13	*S. dysenteriae*	-	+	-	+

## Data Availability

Data sharing is not applicable to this article.

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
