# Peer review of "In Vitro and In Silico Screening and Characterization of Antimicrobial Napin Bioactive Protein in *Brassica juncea* and *Moringa oleifera"

_molecules, 2021, doi:10.3390/molecules26072080_

Round 1

Reviewer 1 Report

There are many places in the manuscript which need language polishing.Please see  examples below.

3. Discussion 
These results based on coagulation activity and protein determination was found to  be correlating with earlier reports on coagulant protein from MO and Mus [11, 12, 29]. On the bases of cell aggregation and antibacterial effect of both BJS and BJP was effective

The top 5 ranked of clusters are taken from Cluspro and run in the surface racer using van  line 565
der waal’s radii as 2, and radius for all the models were taken as 1.4 for surface area cal- line 566 

________________________________________________________________________

Letters in the upper case e.g.Minimum were mixed with letters in the lower case e.g.and Minimum bactericidal activity (MBC)

4.5.2.2. Growth inhibitory activity – determination of Minimum inhibitory activity (MIC)  line 492 and Minimum bactericidal activity (MBC)

______________________________________________________________________

Table 4: Comparative analysis of MOS and BJS extracts towards thirteen different clinical pathogens. __

QUESTION:_Where are the data for MOS and BJS extracts ?????

_Table 5: Please show QUANTITATIVE data rather than qualitative data ___________________________________________________________________

There are references in the reference list with a duplication of the reference number as shown below.

18. 18 Gupta, S., Jain, R., Kachhwaha, S., & Kothari, S. L. (2017). Nutritional and medicinal applications of Moringa oleifera Lam.—Review of current status and future possibilities. Journal of Herbal Medicine. 
19. 19 Sousa, A. M. P., Salles, H. O., de Oliveira, H. D., de Souza, B. B. P., de Lima Cardozo Filho, J., Sifuentes, D. N., & do Egito, A. S. (2020). Mo-HLPs: New flocculating agents identified from Moringa oleifera se

Author Response

Reviewer 1

 The authors were thankful for the reviewer’s valuable comments to enhance the quality of the manuscript and specifically enhance the visibility on the novelty and data profiling of the manuscript based on the reader’s interest

1) There are many places in the manuscript which need language polishing. Please see examples below.

As per the reviewer valuable suggestion the grammatical and typographical errors were corrected throughout the manuscript

2) 3. Discussion

  • These results based on coagulation activity and protein determination was found to be correlating with earlier reports on coagulant protein from MO and Mus [11, 12, 29]. On the bases of cell aggregation and antibacterial effect of both BJS and BJP was effective
  • The top 5 ranked of clusters are taken from Cluspro and run in the surface racer using van line 565 der waal’s radii as 2, and radius for all the models were taken as 1.4 for surface area cal- line 566
  • Letters in the upper case e.g.Minimum were mixed with letters in the lower case e.g.and Minimum bactericidal activity (MBC)

As per the reviewer indicated the typographical errors were corrected throughout the manuscript

3) 4.5.2.2. Growth inhibitory activity – determination of Minimum inhibitory activity (MIC)  line 492 and Minimum bactericidal activity (MBC)

Table 4: Comparative analysis of MOS and BJS extracts towards thirteen different clinical pathogens. __

QUESTION:_Where are the data for MOS and BJS extracts ?????

Based on the reviewer suggestion, the authors would like to provide some clarification on the list of tables applied as 1, 4 and 5, which has been separated follows, if we combine the tables it will be not provide clarification in results description, hence we separated the table to form clarity in the description

  • Antimicrobial activity of plant crude extract and purified coagulant proteins

Table 1. The growth inhibitory activity of Moringa olifera crude extract (MOS), Brassica juncea extract (BJS), Moringa olifera purified protein (MCP) and Brassica juncea purified Protein (MusP) against different clinical pathogens.

  • Analysis on the Antimicrobial resistance of clinical pathogens (isolated from dysentery patients), which was applied in the study

Table 4. Comparative analysis of antibiotics towards thirteen different clinical pathogens.

4) _Table 5: Please show QUANTITATIVE data rather than qualitative data

Based on the reviewer suggestion, the authors would like to provide some clarification on the quantitative data. Which were provide separately in Table 1 and Figure1.a but the Table 5 indicates the overall comparison to understand the combined activity of the porotein

  • Overall view on the combination effect of coagulation and antimicrobial activity. Table 5. Overall view on the combination effect of coagulation (flocculation of artificial clay -kaolin) and antimicrobial activity (growth inhibition) of Moringa olifera (Moringa) and Brassica juncea (Mustard) leaf extracts towards thirteen different clinical pathogens.
  • The growth inhibition data were quantitatively described in the Table 1. The growth inhibitory activity of Moringa olifera crude extract (MOS), Brassica juncea extract (BJS), Moringa olifera purified protein (MCP) and Brassica juncea purified Protein (MusP) against different clinical pathogens. (Antimicrobial activity of plant crude extract and purified coagulant proteins)
  • The coagulation activity were quantitatively described in Figure1.a. Coagulation activity of Moringa oleifera leaf extract (MOS) and Brassica juncea leaf extract (BJS) and coagulant protein (CP) after 90 minutes of sedimentation. The Crude Extract (CEs) were not diluted whereas CPs were 5 times diluted. Kaolin clay without the addition of extract served as a control. Values are expressed as the mean ± standard deviation (n = 3). Different superscripts (a, b, c,) represent significantly different values (P < 0.05).

5) There are references in the reference list with a duplication of the reference number as shown below.

  1. 18 Gupta, S., Jain, R., Kachhwaha, S., & Kothari, S. L. (2017). Nutritional and medicinal applications of Moringa oleifera Lam.—Review of current status and future possibilities. Journal of Herbal Medicine.
  2. 19 Sousa, A. M. P., Salles, H. O., de Oliveira, H. D., de Souza, B. B. P., de Lima Cardozo Filho, J., Sifuentes, D. N., & do Egito, A. S. (2020). Mo-HLPs: New flocculating agents identified from Moringa oleifera se

As per the reviewer valuable suggestion the duplication and typographical error were corrected 

Reviewer 2 Report

The submitted article contains a lot of interesting results, however, some substantial issues are unclear and have to explained.

1) The general idea of the plant extract application for water treatment seems to be unjustified. Do the Authors not afraid that the plant extracts can significantly affect the taste and smell of water after treatment? Could the Authors refer to this issue, especially in the context of drinking water treatment?

2) Why did the Authors assumed that the protein are responsible for coagulation and antibacterial activity? This assuming seems to be rather a priori, in my opinion. Without discussing and justifying it, whole analysis of protein interactions, however valuable, is not entirely relevant to the main topic of this article.

3) What about other extracts components, like e.g. flavones? What kind of them are? What about their antibacterial properties?

4) The isolation of the protein is not clear. How the crude protein fraction (before size separation) was obtained?

5) Why the results from Table 1 is not comment together wtith Table 4 and 5? What is the differences between this analysis?

6) Fig. 1 needs to be supplemented with information about significance of statistical analysis. It refers to Fig. 3, as well.

7) The results in Fig. 3 are puzzling and ambiguous. For example, how the very high values of OD600 for 0.01275 mg/mL in (d) and for .0083 mg/mL in (c) can be explained? 

8) The manuscript should be rewwritten to correct some vocabulary and style errors. For example: please uniform the style of capital/small letter after i. ii. and iii. (lines 72-75); please use small "k" as a prefix for "kilo"; column instead coloum (line 624), etc.

Author Response

Reviewer 2

The submitted article contains a lot of interesting results, however, some substantial issues are unclear and have to explained.

 The authors were thankful for the reviewer’s valuable comments to enhance the quality of the manuscript and specifically enhance the visibility on the novelty and data profiling of the manuscript based on the reader’s interest

1) The general idea of the plant extract application for water treatment seems to be unjustified. Do the Authors not afraid that the plant extracts can significantly affect the taste and smell of water after treatment? Could the Authors refer to this issue, especially in the context of drinking water treatment?

As per the reviewer valuable suggestion the authors have provided a suitable Justification of applying the plant extract for the water treatment are as follows

  • Conventional water purification systems using imported chemicals are prohibitively expensive for many developing countries. Such expensive conventional methods of assuring potable water quality are unsustainable. The mortality rate arising from the use of unsafe water is a major concern for both government and international institutions throughout the world. If the Millennium Development Goals and the targets set by the World Summit on Sustainable Development are to be met, there is a vital need to develop sustainable technologies to treat groundwater for rural livelihoods. The search for locally available low cost materials therefore is inevitable.
  • The most common source of drinking water for the rural people in developing countries is from boreholes (deep wells), shallow wells, springs and rivers. About 37% of population use boreholes as their main source of drinking water and about 26% draw their water from unprotected wells (Staines, 2002). Most of the groundwater is usually consumed without any form of treatment (Pritchard et al., 2007; 2008). Water is a medium for thousands of microorganisms, some of which are disease-causing. Pathogens (e.g. bacteria, viruses, protozoa and helminths) in water cause a variety of diarrhea related diseases such as cholera, typhoid and dysentery. These pathogens are commonly derived from human fecal material. Around 2.5 billion people are without adequate sanitation in the world (UNICEF, 2009). In the rainy season, many pit latrines in the developing world collapse under their own weight due to poor workmanship which further reduces the sanitation coverage. Open defecation in the bush and water bodies is still a popular means of human excreta disposal for rural villagers without access to pit latrines (Lungu et al., 2008). In the rainy season, fecal matter from pit latrines and open sources is washed into water bodies, thereby contaminating the water (Dzwairo et al., 2006). Microbiological water quality from shallow wells in Malawi (with depths, typically, not exceeding 15 m) has been found to be more inferior in the wet season compared to the dry season (Pritchard et al., 2007, 2008).
  • As a solution Natural plant extracts have been applied for water purification for many centuries. Most of these extracts are derived from the seeds, leaves, pieces of bark or sap, roots and fruit extracts of trees and plants. For example, Strychnos potatorum was used as a clarifier between the 14th and 15th centuries BC. Shultz and Okun (1984) together with Sanghi et al. (2006) reported that seeds of the nirmali tree ( potatorum) were used to clarify turbid river water about 4000 years ago in India. It is further reported that in Peru water has been traditionally clarified with the mucilaginous sap of tuna leaves obtained from certain species of cacti. Zea mays was used as a settling agent by sailors in the 16th and 17th centuries.
  • The plant extracts not significantly affect the taste and smell of water after treatment because the smell of the crude extract of Moringa were extracted using 95 % ethanol and  5% aqueous extract was pre-pared using sterile distilled water. Hence the ordure forming volatile compounds were evaporated and removed and further the low concentration of extract were applied hence the taste and order will not be affected
  • The following plant extract were applied for the treatment of water, mainly the application were applied even in the secondary treatment of the sewage water.

Reference

  1. Bodlund I, Sabarigrisan K, Chelliah R, Sankaran K, Rajarao G K. Screening of Coagulant Proteins from Plant Materials in Southern India (Submitted to Water Science and Technology: Water Supply, under review)
  2. Bodlund I, Pavankumar A R, Chelliah R, Sabarigrisan K, Sankaran K, Rajarao G K. Coagulant Proteins Identified in Mustard: A Potential Water Treatment Agent (Submitted to Int. J. Env. Sci. Technol., revised)
  3. Sanghi, R., Bhattacharya, B., Dixit, A., Singh, V., 2006. Ipomoea dasysperma seed gum: an effective natural coagulant for the decolorization of textile dye solutions. Journal of Environmental Management 81 (1), 36–41.
  4. Yongabi, K.A., 2004. Studies on the Potential Use of Medicinal Plants and Macrofungi (Lower Plants) in Water and wastewater Purification, FMENV/ ZERI Research Centre, Abubakar Tafawa Balewa University, Bauchi, Nigeria. (accessed 10.05.09).
  5. Sutherland, J.P., Folkard, G.K., Mtawali, M.A., Grant, W.D., 1994. Moringa oleifera as a natural coagulant. In: 20th WEDC Conference, Affordable Water Supply and Sanitation, Colombo, Sri Lanka.

2) Why did the Authors assumed that the protein are responsible for coagulation and antibacterial activity? This assuming seems to be rather a priori, in my opinion. Without discussing and justifying it, whole analysis of protein interactions, however valuable, is not entirely relevant to the main topic of this article.

Based on the reviewer question, the authors would like to give some clarification on the reason to focus on protein as a bioactive compound towards coagulation and antibacterial activity

  • Previously we performed study on the nutritional and polyphenolic content (secondary metabolite) “Chelliah, R., Ramakrishnan, S., & Antony, U. (2017). Nutritional quality of Moringa oleifera for its bioactivity and antibacterial properties. International Food Research Journal, 24(2), 825”.

  • Further we analyzed the extracted polyphenol for their functional activity such as antimicrobial, antidiabetic, antioxidant, anticancer efficacy. From our previous study the antioxidant and antidiabetic activity was higher but the antimicrobial activity or coagulant activity was not found. Hence we directed our study towards the storage peptides in different parts of the Moringa plant (Seeds, leaves, bark, and root) and it was found that the peptide plays major role in the coagulant and antimicrobial activity based the negative charge possessed by certain small molecular proteins, which act as flocculation peptide with the microbes and (Impurities in the water) turbidities.

Reference

  1. Rockwood, J. L., Anderson, B. G., & Casamatta, D. A. (2013). Potential uses of Moringa oleifera and an examination of antibiotic efficacy conferred by M. oleifera seed and leaf extracts using crude extraction techniques available to underserved indigenous populations. International Journal of Phytotherapy Research, 3(2), 61-71”.
  2. Moulin, M., Mossou, E., Signor, L., Kieffer-Jaquinod, S., Kwaambwa, H. M., Nermark, F., ... & Rennie, A. R. (2019). Towards a molecular understanding of the water purification properties of Moringa seed proteins. Journal of colloid and interface science, 554, 296-304.

3) What about other extracts components, like e.g. flavones? What kind of them are? What about their antibacterial properties?

Based on the reviewer question, the authors would like to give some clarification on the other bioactive compound (secondary metabolites such as Saponins, flavonoids, steroids, glycosides and polyphenols) towards coagulation and antibacterial activity

  • In the current study we tried with the Saponins, flavonoids, steroids, glycosides and polyphenols, which indicated high antioxidant activity but it showed nil effect towards coagulation and antibacterial activity

  • But based on the previous study the secondary metabolite nutritional profiling and its functional activity were described in detail Chelliah, R., Ramakrishnan, S., & Antony, U. (2017). Nutritional quality of Moringa oleifera for its bioactivity and antibacterial properties. International Food Research Journal, 24(2), 825”.
  • Secondary metabolites Saponins, flavonoids, steroids, glycosides and polyphenols were present in leaves and seeds from Madurai and Chennai but Phlobatannins was absent. Tannins and Alkaloids were absent in leaves of both regions. Terpenoids was present in leaves but not in seeds. Polyphenol content was slightly high in Moringa from Madurai (Table 1). Secondary metabolites are activated by enzyme hydrolysis and are used as medications (Kashiwada et al., 2012).
  • This study established that oleifera leaves and seeds contained: tannins, steroids, terpenoids, flavonoids, glycosides, saponins, alkaloids and polyphenols which have been identified by other researchers in various plants and in different parts of plants (Bennett et al., 2003). The findings in this study agree with earlier studies that not all phytochemicals are present in all plant parts and that those present differ according to the type of the extracting solvent used. Flavonoids are also active in reducing high blood pressure (Ayinde et al., 2007). They are many in number as well as strong antioxidants and also found to be effective antimicrobial substances in vitro against a wide array of microorganisms by inhibiting the membrane bound enzymes (Sultana et al., 2007). Tannins are a group of polymeric phenolic substances capable of tanning leather, inactivating and killing microorganisms (Scalbert, 1991).

4) The isolation of the protein is not clear. How the crude protein fraction (before size separation) was obtained?

As per the reviewer valuable suggestion , the authors had provide detailed procedure (steps) , which was performed from extraction to purification of coagulant and antimicrobial protein

Extraction of crude protein (active compounds) from plant leaf

Step 1. Moringa oleifera (MOS) and Mustard Brassica juncea (BJS) leaf were purchased from local shops in Southern India. Preparation of crude extracts from Moringa and Mustard leafs were performed as described earlier

Reference : Sousa, A. M. P., Salles, H. O., de Oliveira, H. D., de Souza, B. B. P., de Lima Cardozo Filho, J., Sifuentes, D. N., & do Egito, A. S. (2020). Mo-HLPs: New flocculating agents identified from Moringa oleifera seeds belong to the hevein-like peptide family. Journal of Proteomics, 217, 103692.

Step 2. Dried MOS and BJS were grounded into fine powder by using mortar and pestle. In order to remove oil from the fine powder, 95 % ethanol was added and the supernatant was separated by centrifugation at 3000 rpm for 10 minutes. The pellet was allowed to air dry and 5% aqueous extract was prepared using sterile distilled water. The obtained soluble fraction is referred as crude extract (CE). 

Step 3. Initially the crude protein of Moringa and Mustard leaf extract were analyzed for the antimicrobial and coagulant activity against thirteen clinical pathogenic strains.

The protein isolation Purification of coagulant and antimicrobial protein

Step 1. Purification of protein based on column chromatography

Based on size-excluding spin column chromatography, a < 30 KDa spin column with < 3 KDa filtrate was subsequently applied to separate the low molecular weight (< 3 KDa) antimicrobial peptides (AMPs).

Step 2. Tris-Tricine SDS-PAGE Electrophoresis and gel-elution

Further based on Tris-Tricine SDS-PAGE (polyacrylamide gel electrophoresis) [21] were applied to separate < 10 KDa peptide band and based on gel elution, < 3 KDa peptide band was eluted and proceed for LC-ESI-TOF-MS/MS (Mass Spectrometry Liquid chromatography-electrospray ionization-quantitative time-of-flight tandem mass spectrometry).

Step 3. Quantification of Protein concentration

The protein content of MOS and BJS crude extracts and coagulant protein was estimated by Lowry’s method [19]. The purity of the coagulant protein was analyzed by Tris-Tricine SDS-PAGE gel analysis according to Laemmli [21] with buffers according to Fling and Gregerson [22]. The gels were stained with Coomassie brilliant blue to visualize the protein.

Step 4. Identification of purified peptides by mass spectrometry

Sequential identification of Peptides by LC-ESI-TOF-MS/MS (Mass Spectrometry Liquid chromatography-electrospray ionization-quantitative time-of-flight tandem mass spectrometry) were analyzed at the National Instrumentation Center for Environmental Management of Seoul National University in Korea, according to an earlier method by Chelliah et al [23]. The peptides were inspected using QS 3.0 software (Applied Biosystems, Seoul, Korea). The ranges of proteins were identified based on 300–3000 m/z values. 

Step 5. SWISS-MODEL: Homology Modelling of Protein Structures

The protein models were spawned by SWISS-MODEL, licensed under the Creative Commons Attribution-Share Alike 4.0 (CC BY-SA 4.0) International License, based on the sequence generated by liquid chromatography mass spectrometry (LCMS). SWISS-MODEL generates theoretical models by automated homology modelling techniques developed by the Computational Structural Biology Group at the Swiss Institute of Bioinformatics (SIB) at the Biozentrum, University of Basel, Switzerland.

5) Why the results from Table 1 is not comment together with Table 4 and 5? What is the differences between this analyses?

Based on the reviewer suggestion, the authors would like to provide some clarification on the list of tables applied as 1, 4 and 5, which has been separated follows, if we combine the tables it will be not provide clarification in results description, hence we separated the table to form clarity in the description

  • Antimicrobial activity of plant crude extract and purified coagulant proteins

Table 1. The growth inhibitory activity of Moringa olifera crude extract (MOS), Brassica juncea extract (BJS), Moringa olifera purified protein (MCP) and Brassica juncea purified Protein (MusP) against different clinical pathogens.

  • Analysis on the Antimicrobial resistance of clinical pathogens (isolated from dysentery patients), which was applied in the study

Table 4. Comparative analysis of antibiotics towards thirteen different clinical pathogens.

  • Overall view on the combination effect of coagulation and antimicrobial activity

Table 5. Overall view on the combination effect of coagulation (flocculation of artificial clay -kaolin) and antimicrobial activity (growth inhibition) of Moringa olifera (Moringa) and Brassica juncea (Mustard) leaf extracts towards thirteen different clinical pathogens.

6) Fig. 1 needs to be supplemented with information about significance of statistical analysis. It refers to Fig. 3, as well.

As per the reviewer suggested the statistical values were incorporated as follows and the figures were edited

Figure1.a. Coagulation activity of Moringa oleifera leaf extract (MOS) and Brassica juncea leaf extract (BJS) and coagulant protein (CP) after 90 minutes of sedimentation. The Crude Extract (CEs) were not diluted whereas CPs were 5 times diluted. Kaolin clay without the addition of extract served as a control. Values are expressed as the mean ± standard deviation (n = 3). Different superscripts (a, b, c,) represent significantly different values (P < 0.05).

Figure 3. Growth inhibition of pathogens based on the different concentrations, further minimum inhibitory concentration was determined in Table 1. a. MOS extract growth inhibitory activity of S. marcescens, b. BJS growth inhibitory activity of S. marcescens, c. MOS extract growth inhibitory activity of S. flexneri, d. BJS growth inhibitory activity of S. flexneri. Values are expressed as the mean ± standard deviation (n = 3).

7) The results in Fig. 3 are puzzling and ambiguous. For example, how the very high values of OD600 for 0.01275 mg/mL in (d) and for .0083 mg/mL in (c) can be explained?

As per the reviewer indicated the graphical error in Figure 3 was rectified

8) The manuscript should be re-written to correct some vocabulary and style errors. For example: please uniform the style of capital/small letter after i. ii. and iii. (lines 72-75); please use small "k" as a prefix for "kilo"; column instead coloum (line 624), etc.

As per the reviewer valuable suggestion the grammatical and typographical errors were corrected throughout the manuscript

Round 2

Reviewer 1 Report

The revised manuscript is acceptable for publication.

Reviewer 2 Report

Thank the Authors for their broad, accurate and informative response.